# Topology-Aware Segmentation Using Discrete Morse Theory

**Xiaoling Hu** *
Stony Brook University

**Yusu Wang**
University of California, San Diego

**Li Fuxin**
Oregon State University

**Dimitris Samaras**
Stony Brook University

**Chao Chen**
Stony Brook University

## ABSTRACT

In the segmentation of fine-scale structures from natural and biomedical images, per-pixel accuracy is not the only metric of concern. Topological correctness, such as vessel connectivity and membrane closure, is crucial for downstream analysis tasks. In this paper, we propose a new approach to train deep image segmentation networks for better topological accuracy. In particular, leveraging the power of discrete Morse theory (DMT), we identify global structures, including 1D skeletons and 2D patches, which are important for topological accuracy. Trained with a novel loss based on these global structures, the network performance is significantly improved especially near topologically challenging locations (such as weak spots of connections and membranes). On diverse datasets, our method achieves superior performance on both the DICE score and topological metrics.

## 1 INTRODUCTION

Segmenting objects while preserving their global structure is a challenging yet important problem. Various methods have been proposed to encourage neural networks to preserve fine details of objects (Long et al., 2015; He et al., 2017; Chen et al., 2014; 2018; 2017). Despite their high per-pixel accuracy, most of them are still prone to structural errors, such as missing small object instances, breaking thin connections, and leaving holes in membranes. These structural errors can significantly damage downstream analysis. For example, in the segmentation of biomedical structures such as membranes and vessels, small pixel errors at a junction will induce significant structure error, leading to catastrophic functional mistakes. See Fig. 1 for an illustration.

Topology is a very global characterization that needs a lot of observations to learn. Any training set is insufficient in teaching the network to correctly reason about topology, especially near challenging spots, e.g., blurred membrane locations or weak vessel connections. A neural network tends to learn from the clean-cut cases and converge quickly. Meanwhile, topologically-challenging locations remain mis-classified, causing structural/topological errors. We note that this issue cannot be alleviated even with more annotated (yet still unbalanced) images.

We propose a novel approach that identifies critical topological structures during training and teaches a neural network to learn from these structures. Our method can produce segmentations with correct topology, i.e., having the same *Betti number* (i.e., number of connected components and handles/tunnels) as the ground truth. Underlying our method is the classic Morse theory (Milnor, 1963), which captures singularities of the gradient vector field of the likelihood function. Intuitively speaking, we treat the likelihood as a terrain function and Morse theory helps us capture terrain structures such as ridges and valleys. See Fig. 1 for an illustration. These structures, composed of 1D and 2D manifold pieces, reveal the topological information captured by the (potentially noisy) likelihood function.

We consider these Morse structures as *topologically critical*; they encompass all potential skeletons of the object. We propose a new loss that identifies these structures and enforce higher penalty along them. This way, we effectively address the sampling bias issue and ensure that the networks predict correctly near these topologically difficult locations. Since the Morse structures are identified based

---

*Email: Xiaoling Hu (xiaolhu@cs.stonybrook.edu).

Figure 1: Illustration of the importance of topological correctness in a neuron image segmentation task and the effectiveness of the proposed DMT-loss. The goal of this task is to segment membranes which partition the image into regions corresponding to neurons. **(a)** an input neuron image with challenging locations (blur regions) highlighted. **(b)** ground truth segmentation of the membranes (dark blue) and the result neuron regions. **(c)** likelihood map of a baseline method without topological guarantee (Ronneberger et al., 2015). **(d)** segmentation results of the baseline method. Small pixel-wise errors lead to broken membranes, resulting in the merging of many neurons into one. **(e)** The topologically critical structure captured by the proposed DMT-loss (based on the likelihood in (c)). **(f)** Our method produces the correct topology and the correct partitioning of neurons.

on the (potentially noisy) likelihood function, they can be both false negatives (a structure can be a true structure but was missed in the segmentation) and false positives (a hallucination of the model and should be removed). Our loss ensures that both kinds of structural mistakes are corrected.

Several technical challenges need to be addressed. First, classical Morse theory was defined for smooth functions on continuous domains. Computing the Morse structures can be expensive and numerically unstable. Furthermore, the entire set of Morse structures may include an excessive amount of structures, a large portion of which can be noisy, irrelevant ones. To address these challenges, we use the discrete version of Morse theory by Forman (1998; 2002). For efficiency purposes, we also use an approximation algorithm to compute 2D Morse structures with almost linear time. The idea is to compute zero dimensional Morse structures of the dual image, which boils down to a minimum spanning tree computation. Finally, we use the theory of persistent homology (Edelsbrunner et al., 2000; Edelsbrunner & Harer, 2010) to prune spurious Morse structures that are not relevant.

Our discrete-Morse-theory based loss, called the *DMT-loss*, can be evaluated efficiently and can effectively train the neural network to achieve high performance in both topological accuracy and per-pixel accuracy. Our method outperforms state-of-the-art methods in multiple topology-relevant metrics (e.g., ARI and VOI) on various 2D and 3D benchmarks. It has superior performance in the Betti number error, which is an exact measurement of the topological fidelity of the segmentation.

**Related work.** Closely related to our method are recent works on persistent-homology-based losses (Hu et al., 2019; Clough et al., 2019). These methods identify a set of critical points of the likelihood function, e.g., saddles and extrema, as topologically critical locations for the neural network to memorize. However, only identifying a *sparse set* of critical points at every epoch is inefficient in terms of training. Instead, our method identifies a much bigger set of critical locations at each epoch, i.e., 1D or 2D Morse skeletons (curves and patches). This is beneficial in both training efficiency and model performance. Extending the critical location sets from points to 1D curves and 2D patches makes it much more efficient in training. Compared with TopoLoss (Hu et al., 2019), we observe a 3-time speedup in practice. Furthermore, by focusing on more critical locations early, our method is more likely to escape poor local minima of the loss landscape. Thus it achieves better topological accuracy than TopoLoss. The shorter training time may also contribute to better stability of the SGD algorithm, and thus better test accuracy (Hardt et al., 2016).

Another topology-aware loss (Mosinska et al., 2018) uses pretrained filters to detect broken connections. However, this method cannot be generalized to unobserved geometry and higher dimensional topology (loops and voids). We also refer to other existing work on topological features and their applications (Adams et al., 2017; Reininghaus et al., 2015; Kusano et al., 2016; Carriere et al., 2017; Ni et al., 2017; Wu et al., 2017). Deep neural networks have also been proposed to learn from topological features directly extracted from data (Hofer et al., 2017; Carrière et al., 2019). Persistent-homology-inspired objective functions have been proposed for graphics (Poulenard et al., 2018) machine learning (Chen et al., 2019; Hofer et al., 2019). Discrete Morse theory has been used to identify skeleton structures from images; e.g., (Delgado-Friedrichs et al., 2015; Robins et al., 2011; Wang et al., 2015). The resulting 1D Morse structure has been used to enhance neural network architecture: e.g., in (Dey et al., 2019) it is used to both pre- and post-process images, while

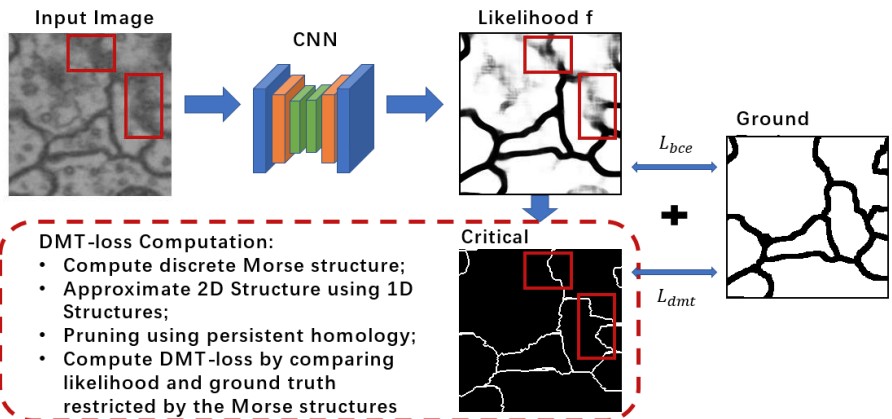

Figure 2: Overview of our method. Topologically critical and error-prune structures are highlighted.

in (Banerjee et al., 2020), the 1D Morse skeleton is used as a topological prior (part of input) for an encoder-decoder deep network for semantic segmentation of microscopic neuroanatomical data. Our work, in contrast, uses Morse structures (beyond 1D) of the output to strengthen the global structural signal more explicitly in an end-to-end training of a network.

Many methods have leveraged the power of deep neural networks (Ronneberger et al., 2015; Long et al., 2015; Badrinarayanan et al., 2017; Ding et al., 2019; Kervadec et al., 2019; Karimi & Salcudean, 2019; Mosinska et al., 2018) for fine-scale structure segmentation. One may also enforce connectivity constraints when postprocessing the likelihood map (Han et al., 2003; Le Guyader & Vese, 2008; Sundaramoorthi & Yezzi, 2007; Ségonne, 2008; Wu et al., 2017; Gao et al., 2013; Vicente et al., 2008; Nowozin & Lampert, 2009; Zeng et al., 2008; Chen et al., 2011; Andres et al., 2011; Stuhmer et al., 2013; Oswald et al., 2014; Estrada et al., 2014). However, when the deep neural network itself is topology-agnostic, the likelihood map may be fundamentally flawed and cannot be salvaged topologically. Specific to neuron image segmentation, some methods (Funke et al., 2017; Turaga et al., 2009; Januszewski et al., 2018; Uzunbas et al., 2016; Ye et al., 2019) directly find neuron regions instead of their boundary/membranes. These methods cannot be generalized to other types of data such as satellite images, retinal images, vessel images, etc.

## 2 METHOD

We propose a novel loss to train a topology-aware network end-to-end. It uses global structures captured by discrete Morse theory (DMT) to discover critical topological structures. In particular, through the language of 1- and 2-stable manifolds, DMT helps identify 1D skeletons or 2D sheets (separating 3D regions) that may be critical for structural accuracy. These Morse structures are used to define a DMT-loss that is essentially the cross-entropy loss constrained to these topologically critical structures. As the training continues, the neural network learns to better predict around these critical structures, and eventually achieves better topological accuracy. Please refer to Fig. 2 for an overview of our method.

### 2.1 MORSE THEORY

Morse theory (Milnor, 1963) identifies topologically critical structures from a likelihood map (Fig. 3(a)). In particular, it views the likelihood as a terrain function (Fig. 3(b)) and extracts its landscape features such as mountain ridges and their high-dimensional counterparts. The broken connection in the likelihood map corresponds to a local dip in the mountain ridge of the terrain in Fig. 3(b) and Fig. 3(c). The bottom of this dip is captured by a so-called saddle point ($S$ in Fig. 3(c)) of the likelihood map. The mountain ridge connected to this bottom point captures the main part of missing pixels. Such "mountain ridges" can be captured by the so-called stable manifold w.r.t. the saddle point using the language of Morse theory. By finding the saddle points and the stable manifold of the saddle points on the likelihood map, we can ensure the model learns to "correctly" handle pixels near these structures. We note that an analogous scenario can also happen with such 1D signals (such as blood vessels) as well as 2D signals (such as membranes of cells) in 3D images – they can also be captured by saddles (of different indices) and their stable manifolds.

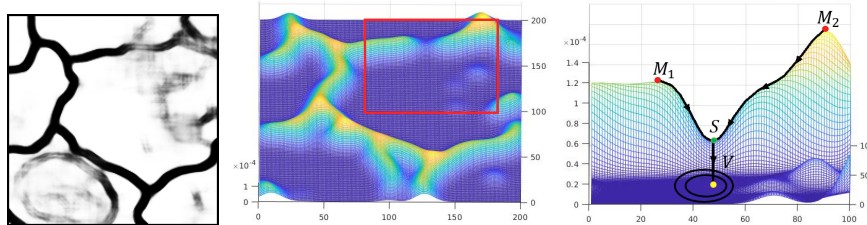

Figure 3: From left to right: **(a)** Likelihood map. **(b)** Density map: the $z$-axis value is the probability of the likelihood map in the left figure. **(c)** Density map for the highlighted region in the middle figure. $M_1$ and $M_2$ are maxima (red dots), $V$ is a minimum (yellow), $S$ is a saddle (green) with its stable manifolds flowing to it from $M_1$ and $M_2$.

In this paper, we focus on the application of segmenting 2D and 3D images. Specifically, suppose we have a smooth function $f : \mathbb{R}^d \to \mathbb{R}$ to model the likelihood (density) map. Given any point $x \in \mathbb{R}^d$, the negative gradient $-\nabla f(x) = -[\frac{\partial f}{\partial x_1}, \frac{\partial f}{\partial x_2}, \ldots, \frac{\partial f}{\partial x_d}]^T$ indicates the steepest descending direction of $f$. A point $x = (x_1, x_2, \ldots, x_k)$ is *critical* if the function gradient at this point vanishes (i.e., $\nabla f(x) = 0$). For a well-behaved function (more formally, called Morse function) defined on $\mathbb{R}^d$, a critical point could be a minimum, a maximum, or $d-1$ types of saddle points. See Fig. 3(c) for an example. For $d = 2$, there is only one saddle point type. For $d = 3$, there are two saddle point types, referred to as index-1 and index-2 saddles. Formally, when taking the eigen-values of the Hessian matrix at a critical point, its index is equal to the number of negative eigenvalues.

Intuitively, imagine we put a drop of water on the graph of $f$ (i.e, the terrain in Fig. 3(b)) at the lift of $x$ onto this terrain, then $-\nabla f(x)$ indicates the direction along which the water will flow down. If we track the trajectory of this water drop as it flows down, this gives rise to a so-called *integral line* (a flow line). Such flow lines can only start and end at critical points[1], where the gradient vanishes.

The stable manifold $\mathsf{S}(p)$ of a critical point $p$ is defined as the collection of points whose flow line ends at $p$. For a 2D function $f : \mathbb{R}^2 \to \mathbb{R}$, for a saddle $q$, its stable manifold $\mathsf{S}(q)$ starts from local maxima (mountain peaks in the terrain) and ends at $q$, tracing out the mountain ridges separating different valleys (Fig. 3(c)). The stable manifold $\mathsf{S}(p)$ of a minimum $p$, on the other hand, corresponds to the entire valley around this minimum $p$. See the valley point $V$ and its corresponding stable field in Fig. 3(c). For a 3D function $f : \mathbb{R}^3 \to \mathbb{R}$, the stable manifold w.r.t. an index-2 saddle connects mountain peaks to saddles, tracing 1D mountain ridges as in the case of a 2D function. The stable manifold w.r.t. an index-1 saddle $q$ consists of flow lines starting at index-2 saddles and ending at $q$. Their union, called the 2-stable manifold of $f$, consists of a collection of 2-manifold pieces.

These stable manifolds indicate important topological structures (graph-like or sheet-like) based on the likelihood of the current neural network. Using these structures, we will propose a novel loss (Sec. 2.3) to improve the topological awareness of the model. In practice, for images, we will leverage the discrete version of Morse theory for both numerical stability and easier simplification.

**Discrete Morse theory.** Due to space limitations, we briefly explain discrete Morse theory, leaving technical details to Appendix A.2.1. We view a $d$-dimensional image, $d = 2$ or $3$, as a $d$-dimensional cubical complex, meaning it consists of 0-, 1-, 2- and 3-dimensional cells corresponding to vertices (pixels/voxels), edges, squares, and cubes as its building blocks. Discrete Morse theory (DMT), originally introduced in (Forman, 1998; 2002), is a combinatorial version of Morse theory for general cell complexes. In this setting, the analog of a gradient vector is a pair of adjacent cells, called discrete gradient vectors. The analog of an integral line (flow line) is a sequence of such cell-pairs (discrete gradient vectors), forming a so-called *V-path*. Critical points correspond to critical cells which do not participate in any discrete gradient vectors. A minimum, an index-1 saddle, an index-2 saddle and a maximum for a 3D domain intuitively correspond to a critical vertex, a critical edge, a critical square and a critical cube respectively. A 1-stable manifold in 2D will correspond to a V-path, i.e., a sequence of cells, connecting a critical square (a maximum) and a critical edge (a saddle). See Fig. 3(c) for an illustration. In 3D, it will be a V-path connecting a critical cube and a critical square.

---

[1]More precisely, flow lines only tend to critical points in the limit and never reach them.

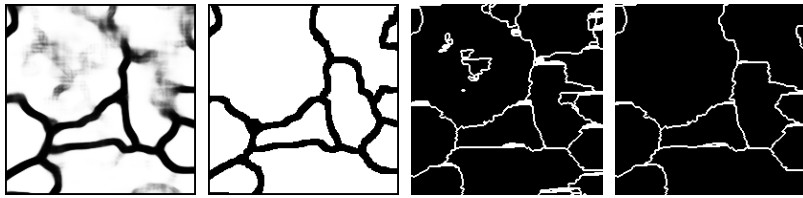

Figure 4: From left to right: **(a)** Sample likelihood map, **(b)** Ground truth, **(c)** improperly pruned structures and **(d)** properly pruned structures.

## 2.2 SIMPLIFICATION AND COMPUTATION

In this section, we describe how we extract discrete Morse structures corresponding to the 1-stable and 2-stable manifolds in the continuous analog. First, we prune unnecessary Morse structures, based on the theory of persistent homology (Edelsbrunner et al., 2000; Edelsbrunner & Harer, 2010). Second, we approximate the 2-stable manifold structures using 0-stable manifolds of the dual to achieve high efficiency in practice, because it is rather involved to compute based on the original definition.

**Persistence-based structure pruning.** While Morse structures reveal important structural information, they can be sensitive to noise. Without proper pruning, there can be an excessive amount of Morse structures, many of which are spurious and not relevant to the true signal. See Fig. 4(c) for an example. Similar to previous approaches e.g., (Sousbie, 2011; Delgado-Friedrichs et al., 2015; Wang et al., 2015), we will prune these structures using persistent homology.

Persistent homology is one of the most important developments in the field of topological data analysis in the past two decades (Edelsbrunner & Harer, 2010; Edelsbrunner et al., 2000). Intuitively speaking, we grow the complex by starting from the empty set and gradually include more and more cells using a decreasing threshold. Through this course, new topological features can be created upon adding a critical cell, and sometimes a feature can be destroyed upon adding another critical cell. The persistence algorithm (Edelsbrunner et al., 2000) will pair up these critical cells; that is, its output is a set of critical cell pairs, where each pair captures the birth and death of topological features during this evolution. The persistence of a pair is defined as the difference of function values of the two critical cells, intuitively measuring how long the topological feature lives in term of $f$.

Using persistence, we can prune critical cells that are less topologically salient, and thus their corresponding Morse structures. Recall each 1- and 2-stable Morse structure is constituted by V-paths flowing into a critical cell (corresponding to a saddle in the continuous setting). We then use the persistence associated with this critical cell to determine the saliency of the corresponding Morse structure. If the persistence is below a certain threshold $\epsilon$, we prune the corresponding Morse structure via an operation called *Morse cancellation* (more details are in the Appendix A.2.2).[2] See Fig. 4(d) for example Morse structures after pruning. We denote by $\mathcal{S}_1(\epsilon)$ and $\mathcal{S}_2(\epsilon)$ the remaining sets of 1- and 2-stable manifolds after pruning. We'll use these Morse structures to define the loss (Sec. 2.3).

**Computation.** We need an efficient algorithm to compute $\mathcal{S}_1(\epsilon)$ and $\mathcal{S}_2(\epsilon)$ from a given likelihood $f$, because this computation needs to be carried out at each epoch. It is significantly more involved to define and compute $\mathcal{S}_2(\epsilon)$ in the discrete Morse setting (Delgado-Friedrichs et al., 2015). Furthermore, this also requires the computation of persistent homology up to 2-dimensions, which takes time $T = O(n^\omega)$ (where $\omega \approx 2.37$ is the exponent in the matrix multiplication time, i.e., the time to multiply two $n \times n$ matrices). To this end, we propose to approximate $\mathcal{S}_2(\epsilon)$ by $\widehat{\mathcal{S}}_2(\epsilon)$ (more details can be found in Appendix A.2.3) which intuitively comes from the "boundary" of the stable manifold for minima. Note that in the smooth case for a function $f : \mathbb{R}^3 \to \mathbb{R}$, the closure of 2-stable manifolds exactly corresponds to the 2D sheets on the boundary of stable manifolds of minima. [3] This is both conceptually clear and also avoids the costly persistence computation. In particular, given a minimum $q$ with persistence greater than the pruning threshold $\epsilon$, the collections of V-paths ending at the minimum $q$ form a spanning tree $T_q$. In fact, consider all minima $\{q_1, \ldots, q_\ell\}$ with persistence at least $\epsilon$. $\{T_{q_i}\}$ form a maximum spanning forest among all edges with persistence value smaller than $\epsilon$ (Bauer et al., 2012; Dey et al., 2018). Hence it can be computed easily in $O(n \log n)$ time, where $n$ is the image size.

---

[2]Technically, not all spurious structures can be pruned/cancelled. But in practice, most of them can.

[3]This however is not always true for the discrete Morse setting.

We then take all the edges incident to nodes from different trees. The dual of these edges, denoted as $\widehat{\mathcal{S}}_2(\epsilon)$, serves as the "boundaries" separating different spanning trees (representing stable manifolds to different minima with persistence $\geq \epsilon$). See Fig. 5. Overall, the computation of $\widehat{\mathcal{S}}_2(\epsilon)$ takes only $O(n \log n)$ by a maximum spanning tree algorithm.

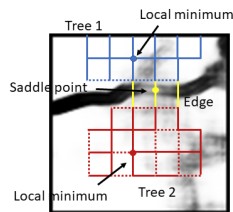

Figure 5: Spanning tree illustration.

As for $\mathcal{S}_1(\epsilon)$, we use a simplified algorithm of (Dey et al., 2018), which can compute $\mathcal{S}_1(\epsilon)$ in $O(n \log n)$ time for a 2D image, in which $n$ is the image size. For a 3D image, the time is $O(n \log n + T)$, where $T = O(n^\omega)$ is the time to compute persistent homology, where $\omega \approx 2.37$ is the exponent in matrix multiplication time.

### 2.3 THE DMT-BASED LOSS FUNCTION AND TRAINING DETAILS

Our loss has two terms, the cross-entropy term, $L_{bce}$ and the DMT-loss, $L_{dmt}$: $L(f, g) = L_{bce}(f, g) + \beta L_{dmt}(f, g)$, in which $f$ is the likelihood, $g$ is the ground truth, and $\beta$ is the weight of $L_{dmt}$. Here we focus on one single image, while the actual loss is aggregated over the whole training set.

The DMT-loss enforces the correctness of the topologically challenging locations discovered by our algorithm. These locations are pixels of the (approximation of) 1- and 2-stable manifolds $\mathcal{S}_1(\epsilon)$ and $\widehat{\mathcal{S}}_2(\epsilon)$ of the likelihood, $f$ (Fig. 1(e)). Denote by $\mathcal{M}_f$ a binary mask of the union of pixels of all Morse structures in $\mathcal{S}_1(\epsilon) \cup \widehat{\mathcal{S}}_2(\epsilon)$. We want to enforce these locations to be correctly segmented. We use the cross-entropy between the likelihood map $f$ and ground truth $g$ restricted to the Morse structures, formally, $L_{dmt}(f, g) = L_{bce}(f \circ \mathcal{M}_f, g \circ \mathcal{M}_f)$, in which $\circ$ is the Hadamard product.

**Different topological error types.** Recall that the Morse structures are computed over the potentially noisy likelihood function of a neural network, which can help identify two types of structural errors: (1) **false negative**: a true structure that is incomplete in the segmentation, but can be visible in the Morse structures. This types of false negatives (broken connections, holes in membrane) can be restored as the additional cross-entropy loss near the Morse structures will force the network to increase its likelihood value on these structures. (2) **false positive**: phantom structures hallucinated by the network when they do not exist (spurious branches, membrane pieces). These errors can be eliminated as the extra cross entropy loss on these structures will force the network to decrease the likelihood values along these structures. We provide a more thorough discussion in the Appendix A.1.2.

**Differentiability.** We note that the Morse structures are recomputed at every epoch. The structures, as well as their mask $\mathcal{M}_f$, may change with $f$. However, the change is not continuous; the output of the discrete Morse algorithm is a combinatorial solution that does not change continuously with $f$. Instead, it only changes at singularities, i.e., when the function values of $f$ at different pixels/voxels are the same. In other words, for a general $f$, the likelihood function is real-valued, so it is unlikely two pixels share the exact same value. In case that they are, the persistence homology algorithm by default will break the tie and choose one as critical. The mask $\mathcal{M}_f$ remains a constant within a small neighborhood of current $f$. Therefore, the gradient of $L_{dmt}$ exists and can be computed naturally.

**Training details.** Although our method is architecture-agnostic, for 2D datasets, we select an architecture driven by a 2D U-net (Ronneberger et al., 2015); for 3D datasets, we select an architecture inspired by a 3D U-Net (Çiçek et al., 2016). Both U-Net and 3D U-Net were originally designed for neuron segmentation tasks, capturing the fine-structures of images. In practice, we first pretrain the network with only the cross-entropy loss, and then train the network with the combined loss.

### 3 EXPERIMENTS ON 2D AND 3D DATASETS

**Experiments on 2D dataset.** Six natural and biomedical 2D datasets are used: **ISBI12** (Arganda-Carreras et al., 2015), **ISBI13** (Arganda-Carreras et al., 2013), **CREMI**, **CrackTree** (Zou et al., 2012), **Road** (Mnih, 2013) and **DRIVE** (Staal et al., 2004). More details about the datasets are included in Appendix A.3.1. For all the experiments, we use a 3-fold cross-validation to tune hyperparameters for both the proposed method and other baselines, and report the mean performance over the validation set. This also holds for 3D experiments.

**Evaluation metrics.** We use five different evaluation metrics: **Pixel-wise accuracy**, **DICE score**, **ARI**, **VOI** and the most important one is **Betti number error**, which directly compares the topology

Table 1: Quantitative results for different models on several 2D image datasets

| Method | Accuracy | DICE | ARI | VOI | Betti Error |
|---|---|---|---|---|---|
| | | ISBI13 | | | |
| DIVE | $0.9642 \pm 0.0018$ | $0.9658 \pm 0.0020$ | $0.6923 \pm 0.0134$ | $2.790 \pm 0.025$ | $3.875 \pm 0.326$ |
| U-Net | $0.9631 \pm 0.0024$ | $0.9649 \pm 0.0057$ | $0.7031 \pm 0.0256$ | $2.583 \pm 0.078$ | $3.463 \pm 0.435$ |
| Mosin. | $0.9578 \pm 0.0029$ | $0.9623 \pm 0.0047$ | $0.7483 \pm 0.0367$ | $1.534 \pm 0.063$ | $2.952 \pm 0.379$ |
| TopoLoss | $0.9569 \pm 0.0031$ | $\mathbf{0.9689 \pm 0.0026}$ | $0.8064 \pm 0.0112$ | $1.436 \pm 0.008$ | $\mathbf{1.253 \pm 0.172}$ |
| DMT | $0.9625 \pm 0.0027$ | $\mathbf{0.9712 \pm 0.0047}$ | $\mathbf{0.8289 \pm 0.0189}$ | $\mathbf{1.176 \pm 0.052}$ | $\mathbf{1.102 \pm 0.203}$ |
| | | CREMI | | | |
| DIVE | $0.9498 \pm 0.0029$ | $0.9542 \pm 0.0037$ | $0.6532 \pm 0.0247$ | $2.513 \pm 0.047$ | $4.378 \pm 0.152$ |
| U-Net | $0.9468 \pm 0.0048$ | $0.9523 \pm 0.0049$ | $0.6723 \pm 0.0312$ | $2.346 \pm 0.105$ | $3.016 \pm 0.253$ |
| Mosin. | $0.9467 \pm 0.0058$ | $0.9489 \pm 0.0053$ | $0.7853 \pm 0.0281$ | $1.623 \pm 0.083$ | $1.973 \pm 0.310$ |
| TopoLoss | $0.9456 \pm 0.0053$ | $0.9596 \pm 0.0029$ | $0.8083 \pm 0.0104$ | $1.462 \pm 0.028$ | $\mathbf{1.113 \pm 0.224}$ |
| DMT | $0.9475 \pm 0.0031$ | $\mathbf{0.9653 \pm 0.0019}$ | $\mathbf{0.8203 \pm 0.0147}$ | $\mathbf{1.089 \pm 0.061}$ | $\mathbf{0.982 \pm 0.179}$ |
| | | CrackTree | | | |
| DIVE | $0.9854 \pm 0.0052$ | $0.6530 \pm 0.0017$ | $0.8634 \pm 0.0376$ | $1.570 \pm 0.078$ | $1.576 \pm 0.287$ |
| U-Net | $0.9821 \pm 0.0097$ | $0.6491 \pm 0.0029$ | $0.8749 \pm 0.0421$ | $1.625 \pm 0.104$ | $1.785 \pm 0.303$ |
| Mosin. | $0.9833 \pm 0.0067$ | $0.6527 \pm 0.0010$ | $0.8897 \pm 0.0201$ | $1.113 \pm 0.057$ | $1.045 \pm 0.214$ |
| TopoLoss | $0.9826 \pm 0.0084$ | $0.6732 \pm 0.0041$ | $\mathbf{0.9291 \pm 0.0123}$ | $0.997 \pm 0.011$ | $\mathbf{0.672 \pm 0.176}$ |
| DMT | $0.9842 \pm 0.0041$ | $\mathbf{0.6811 \pm 0.0047}$ | $\mathbf{0.9307 \pm 0.0172}$ | $\mathbf{0.901 \pm 0.081}$ | $\mathbf{0.518 \pm 0.189}$ |
| | | Road | | | |
| DIVE | $0.9734 \pm 0.0077$ | $0.6743 \pm 0.0051$ | $0.8201 \pm 0.0128$ | $2.368 \pm 0.203$ | $3.598 \pm 0.783$ |
| U-Net | $0.9786 \pm 0.0052$ | $0.6612 \pm 0.0016$ | $0.8189 \pm 0.0097$ | $2.249 \pm 0.175$ | $3.439 \pm 0.621$ |
| Mosin. | $0.9754 \pm 0.0043$ | $0.6673 \pm 0.0044$ | $0.8456 \pm 0.0174$ | $1.457 \pm 0.096$ | $2.781 \pm 0.237$ |
| TopoLoss | $0.9728 \pm 0.0063$ | $0.6903 \pm 0.0038$ | $0.8671 \pm 0.0068$ | $1.234 \pm 0.037$ | $\mathbf{1.275 \pm 0.192}$ |
| DMT | $0.9744 \pm 0.0049$ | $\mathbf{0.7056 \pm 0.0022}$ | $\mathbf{0.8819 \pm 0.0104}$ | $\mathbf{1.092 \pm 0.129}$ | $0.995 \pm 0.301$ |

(number of handles/voids) between the segmentation and the ground truth. More details about the evaluation metrics are provided in Appendix A.3.2. The last three metrics are topology-aware.

**Baselines.** **DIVE** (Fakhry et al., 2016) is a popular neural network that predicts the probability of a pixel being a membrane (border) pixel or not. **U-Net** (Ronneberger et al., 2015) is one of the most powerful image segmentation methods trained with cross-entropy loss. **Mosin.** (Mosinska et al., 2018) uses a topology-aware loss based on the response of selected filters from a pretrained CNN. **TopoLoss** (Hu et al., 2019) proposes a novel topological loss to learn to segment with correct topology. For all methods, we generate segmentations by thresholding the predicted likelihood maps at 0.5, and this also holds for 3D experiments.

**Quantitative and qualitative results.** Table 1 shows quantitative results for four 2D image datasets, ISBI13, CREMI, CrackTree and Road. The results are highlighted when they are significantly better, and statistical significance is determined by t-tests. Results for ISBI12 and DRIVE are included in Appendix A.3.3. The DMT-loss outperforms others in both DICE score and topological accuracy (ARI, VOI and Betti Error). Please note that here the backbone of TopoLoss is the same as in (Hu et al., 2019), a heavily engineered network. The performance of TopoLoss will be worse if we implement it using the same U-Net backbone as DMT-Loss. Comparisons with the same backbones can be found in Appendix A.3.4.

Fig. 6 shows qualitative results. Our method correctly segments fine structures such as membranes, roads and vessels. Our loss is a weighted combination of the cross entropy and DMT-losses. When $\beta = 0$, the proposed method degrades to a standard U-Net. The performance improvement over all datasets (U-Net and DMT line in Table 1) demonstrates that our DMT-loss is helping the deep neural nets to learn a better structural segmentation.

**Experiments on 3D datasets.** We use three different biomedical 3D datasets: **ISBI13**, **CREMI** and **3Dircadb** (Soler et al., 2010). **ISBI13** and **CREMI**, which have been discussed above, are originally 3D datasets, can be used in both 2D and 3D evaluations. We also evaluate our method on the open dataset **3Dircadb**, which contains 20 enhanced CT volumes with artery annotations.

**Evaluation metrics.** We use similar evaluation metrics as the 2D part. Note that, in term of 2D images, for ARI and VOI, we compute the numbers for each slice and then average the numbers for different slices as the final performance; for 3D images, we compute the performance for the whole

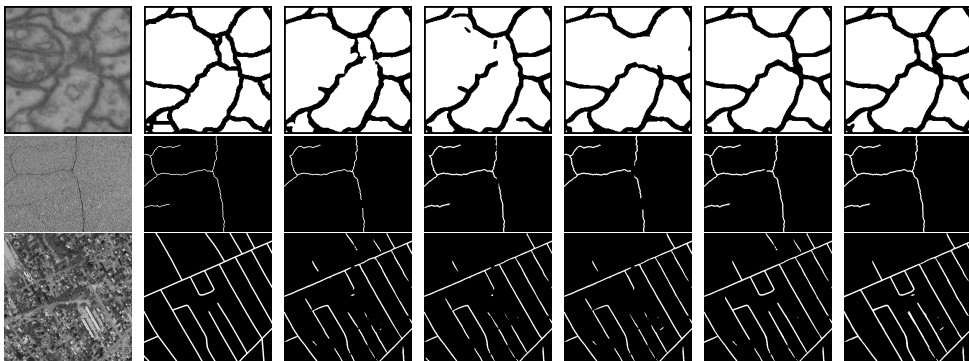

Figure 6: Qualitative results of the proposed method compared to other models. From left to right: sample images, ground truth, results for **DIVE**, **U-Net**, **Mosin.**, **TopoLoss** and our proposed **DMT**.

Table 2: Quantitative results for different models on several 3D datasets

| Method | Accuracy | DICE | ARI | VOI | Betti Error |
|---|---|---|---|---|---|
| | | ISBI13 | | | |
| 3D DIVE | $0.9723 \pm 0.0021$ | $0.9681 \pm 0.0043$ | $0.8719 \pm 0.0189$ | $1.208 \pm 0.149$ | $2.375 \pm 0.419$ |
| 3D U-Net | $0.9746 \pm 0.0025$ | $0.9701 \pm 0.0012$ | $\mathbf{0.8956 \pm 0.0391}$ | $1.123 \pm 0.091$ | $1.954 \pm 0.585$ |
| MALA | $0.9701 \pm 0.0018$ | $0.9699 \pm 0.0013$ | $\mathbf{0.8945 \pm 0.0481}$ | $0.901 \pm 0.106$ | $1.103 \pm 0.207$ |
| 3D TopoLoss | $0.9689 \pm 0.0031$ | $0.9752 \pm 0.0045$ | $\mathbf{0.9043 \pm 0.0283}$ | $0.792 \pm 0.086$ | $0.972 \pm 0.245$ |
| DMT | $0.9701 \pm 0.0026$ | $\mathbf{0.9803 \pm 0.0019}$ | $\mathbf{0.9149 \pm 0.0217}$ | $\mathbf{0.634 \pm 0.086}$ | $\mathbf{0.812 \pm 0.134}$ |
| | | CREMI | | | |
| 3D DIVE | $0.9503 \pm 0.0061$ | $0.9641 \pm 0.0011$ | $0.8514 \pm 0.0387$ | $1.219 \pm 0.103$ | $2.674 \pm 0.473$ |
| 3D U-Net | $0.9547 \pm 0.0038$ | $0.9618 \pm 0.0026$ | $0.8322 \pm 0.0315$ | $1.416 \pm 0.097$ | $2.313 \pm 0.501$ |
| MALA | $0.9472 \pm 0.0027$ | $0.9583 \pm 0.0023$ | $0.8713 \pm 0.0286$ | $1.109 \pm 0.093$ | $1.114 \pm 0.309$ |
| 3D TopoLoss | $0.9523 \pm 0.0043$ | $0.9672 \pm 0.0010$ | $0.8726 \pm 0.0194$ | $1.044 \pm 0.128$ | $1.076 \pm 0.206$ |
| DMT | $0.9529 \pm 0.0031$ | $\mathbf{0.9731 \pm 0.0045}$ | $\mathbf{0.9013 \pm 0.0202}$ | $\mathbf{0.891 \pm 0.099}$ | $\mathbf{0.726 \pm 0.187}$ |
| | | 3Dircadb | | | |
| 3D DIVE | $0.9618 \pm 0.0054$ | $0.6097 \pm 0.0034$ | / | / | $4.571 \pm 0.505$ |
| 3D U-Net | $0.9632 \pm 0.0009$ | $0.5898 \pm 0.0025$ | / | / | $4.131 \pm 0.483$ |
| MALA | $0.9546 \pm 0.0033$ | $0.5719 \pm 0.0043$ | / | / | $2.982 \pm 0.105$ |
| 3D TopoLoss | $0.9561 \pm 0.0019$ | $0.6138 \pm 0.0029$ | / | / | $2.245 \pm 0.255$ |
| DMT | $0.9587 \pm 0.0023$ | $\mathbf{0.6257 \pm 0.0021}$ | / | / | $\mathbf{1.415 \pm 0.305}$ |

volume. For 2D images, we compute the 1D Betti number (number of holes) to obtain the Betti Error; while for 3D images, we compute the 2D Betti number (number of voids) to obtain the Betti Error.

**Baselines.** **3D DIVE** (Zeng et al., 2017), **3D U-Net** (Çiçek et al., 2016), **3D TopoLoss** (Hu et al., 2019) are the 3D versions for **DIVE**, **U-Net** and **TopoLoss**. **MALA** (Funke et al., 2017) trains the U-Net using a new structured loss function.

**Quantitative and qualitative results.** Table 2 shows the quantitative results for three different 3D image datasets, ISBI13, CREMI and 3Dircadb. Our method outperforms existing methods in topological accuracy (in all three topology-aware metrics), which demonstrates the effectiveness of the proposed method. More qualitative results for 3D cases are included in Appendix A.3.5.

**The benefit of the proposed DMT-loss.** Instead of capturing isolated critical points in TopoLoss (Hu et al., 2019), the proposed DMT-loss captures the whole V-path as critical structures. Taking the patch in Fig. 4(a) as an example, TopoLoss identifies $\approx 80$ isolated critical pixels for further training, whereas the critical structures captured by the DMT-loss contain $\approx 1000$ critical pixels (Fig. 4(d)). We compare the efficiency of DMT-loss and TopoLoss using the same backbone network, evaluated on the CREMI 2D dataset. Both methods start from a reasonable pre-trained likelihood map. TopoLoss achieves 1.113 (Betti Error), taking $\approx$3h to converge; while DMT-loss achieves 0.956 (Betti Error), taking $\approx$1.2h to converge (the standard U-Net takes $\approx$0.5h instead). Aside from converging faster, the DMT-loss is also less likely to converge to low-quality local minima. We hypothesize that the loss landscape of the topological loss will have more local minima than that of the DMT-loss, even though the global minima of both landscapes may have the same quality.

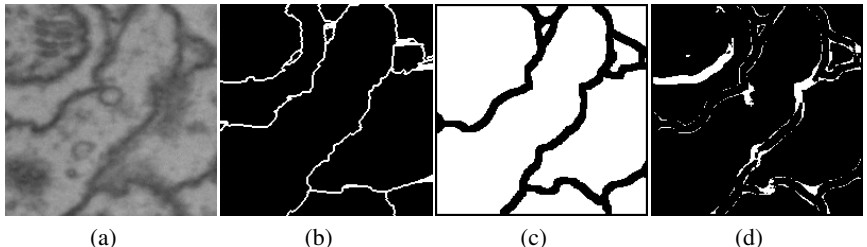

(a)        (b)        (c)        (d)

Figure 7: Illustration of comparison between the proposed DMT-loss and a simple reweighted cross entropy loss. Please refer to Fig. 10(b) for the likelihood map of a baseline method without topological guarantee. **(a)** an input neuron image. **(b)** The topologically critical structures from the likelihood, captured by the proposed discrete Morse algorithm. These structures will be used in the DMT-Loss. **(c)** ground truth. **(d)** The FP/FN pixels identified by simple re-weighting cross entropy loss.

**Ablation study for persistence threshold $\epsilon$.** As illustrated in Fig. 4, different persistence thresholds $\epsilon$ will identify different critical structures. The ablation experiment is also conducted on the CREMI 2D dataset. When $\epsilon = 0.2$ (See Fig. 4(d)), the proposed DMT-loss achieves the best performance 0.982 (Betti Error). When $\epsilon = 0.1$ and $\epsilon = 0.3$, the performance drops to 1.206 and 2.105 (both in Betti Error), respectively. This makes sense for the following reasons: 1) for $\epsilon = 0.1$, the DMT-loss captures lots of unnecessary structures which mislead the neural networks; 2) for $\epsilon = 0.3$, the DMT-loss misses lots of important critical structures, making the performance drop significantly. The $\epsilon$ is chosen via cross-validation. The ablation study for balanced term $\beta$ is provided in Appendix A.3.6.

**Comparison with other simpler choices.** The proposed method essentially highlights geometrically rich locations. To demonstrate the effectiveness of the proposed method, we also compare with two baselines: canny edge detection, and ridge detection, which achieve 2.971 and 2.507 in terms of Betti Error respectively, much worse than our results (Betti Error: 0.982). Although our focus is the Betti error, we also report per-pixel accuracy for reference (See Table 5 in Appendix for details). From the results we observe that the baseline models could not solve topological errors, even though they achieve high per-pixel accuracy. Without a persistent-homology based pruning, these baselines generate too many noisy structures, and thus are not as effective as DMT-loss.

**Robustness of the proposed method.** We run another ablation study on images corrupted with Gaussian noise. The experiment is also conducted on the CREMI 2D dataset. From the results (See Table 6 in Appendix for details), the DMT-loss is fairly robust and maintains good performance even with high noise levels. The reason is that the Morse structures are computed on the likelihood map, which is already robust to noise.

**Comparison with reweighted cross entropy loss.** We run an additional ablation study to compare with a baseline of reweighting the FP and FN pixels in the cross-entropy loss (Reweighting CE). The weights of the FP/FN pixels are hyperparameters tuned via cross-validation. The reweighting CE strategy achieves 2.753 in terms of Betti Error (on CREMI 2D data), and the DMT-loss is better than this baseline. The reason is that the DMT-loss specifically penalizes FP and FN pixels which are topology-critical. Meanwhile, reweighting CE adds weights to FP/FN pixels without discrimination. A majority of these misclassified pixels are not topology-critical. They are near the boundary of the foreground. Please see Fig. 7 for an illustration.

## 4 CONCLUSION

In this paper, we proposed a new DMT-loss to train deep image segmentation neural networks for better topological accuracy. With the power of discrete Morse theory (DMT), we could identify 1D skeletons and 2D patches which are important for topological accuracy. Trained with the new loss based on these global structures, the networks perform significantly better, especially near topologically challenging locations (such as weak spots of connections and membranes).

**Acknowledgements.** The research of Xiaoling Hu and Chao Chen is partially supported by NSF IIS-1909038. The research of Li Fuxin is partially supported by NSF IIS-1911232. The research of Yusu Wang is partially supported by NSF under grants CCF-1733798, RI-1815697, and by NIH under grants R01EB022899, RF1MH125317. Dimitris Samaras is partially supported by a gift from Adobe, the Partner University Fund, the NSF IUCRC for Visual and Decision Informatics and the SUNY2020 Infrastructure Transportation Security Center.

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

## A APPENDIX

In Sec. A.1, we will discuss different types of topological errors. In Sec. A.2, we will provide more details for the method part. In Sec. A.3, more details and results for the experiments will be provided.

### A.1 DIFFERENT TYPES OF TOPOLOGICAL ERRORS

#### A.1.1 ILLUSTRATION OF 3D TOPOLOGICAL ERRORS

In Sec.2.1 of the main paper, we have already introduced discrete Morse theory with a 2D example. Here, we would like to illustrate 3D topological errors with 3D examples.

Fig. 8 and Fig. 9 illustrate two different types of topological errors for 3D data. Fig.8 illustrates an index-1 topological error for 3D synthetic data. 3D EM/neuron has the same type of topological error as the synthetic data. Fig.9 illustrates index-2 topological error for 3D vessel data.

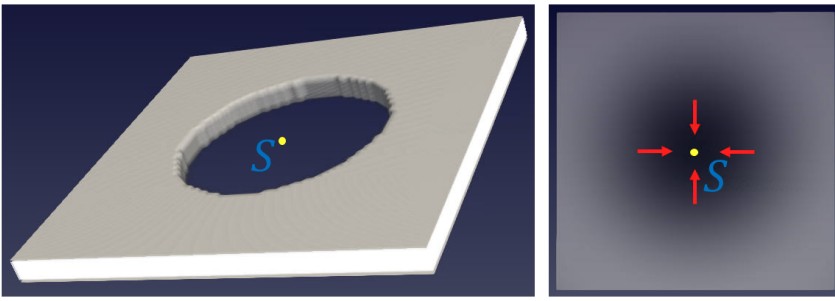

Figure 8: Illustration of an index-1 topological error (3D hole in the middle) for a 3D synthetic data. The ground truth is a complete sheet without the hole. We intentionally weaken the likelihood function in the middle. So the segmentation has a hole in the middle. **Left**: 3D segmentation result. $S$ is a saddle point of the likelihood function. Its Hessian has 1 negative and 2 positive eigenvalues. The stable manifold of the saddle point $S$ is a 2D plane going through the saddle point and cutting the segmentation into two thin slices. **Right**: the likelihood function visualized on the 2D stable manifold of $S$. Red arrows illustrate how different $V$-paths (streamlines of negative gradient) flow to the saddle $S$.

#### A.1.2 FALSE NEGATIVE AND FALSE POSITIVE ERRORS

In the main paper, we have mentioned that the proposed DMT-loss can capture and fix two different types of topological errors: false negative and false positive. We illustrate these two types in Fig. 10. The two highlighted red rectangles represent the two types of topological errors: 1) The red rectangle on the right represents a sample of false negative error; part of the membrane structure is missing, due to a blurred region near the membrane. 2) The red rectangle on the left represents a sample of false positive error. In this specific case, it is caused by mitochondria which are not the boundary of neurons.

In summary, with the help of the proposed DMT-loss, we can identify both these two types of topological errors, and then force the network to increase/decrease its likelihood value on these structures to correctly segment the images with with correct topology.

### A.2 ADDITIONAL DETAILS ON THE METHOD

#### A.2.1 DISCRETE MORSE THEORY

We view a $d$D image, $d = 2$ or $3$, as a $d$-dimensional cubical complex, meaning it consists of 0-, 1-, 2- and 3-dimensional cells corresponding to vertices, edges, squares, and voxels (cubes) as its building blocks.

Discrete Morse theory (DMT), originally introduced in (Forman, 1998; 2002), is a combinatorial version of Morse theory for general cell complexes. There are many beautiful results established for

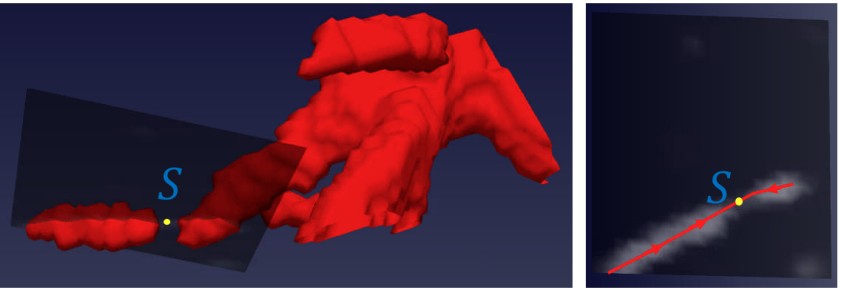

Figure 9: Illustration of an index-2 topological error from a 3D vessel image. The vessel segmentation has a broken connection near bottom-left of the Left image. **Left**: part of the 3D segmentation result. A The saddle point $S$ corresponds to a broken connection. The Hessian of the likelihood at the saddle point has 2 negative and 1 positive eigenvalues. **Right**: One slice of the 3D likelihood map passing the saddle point. The saddle point (yellow) and its 1-stable manifold (red) are also drawn.

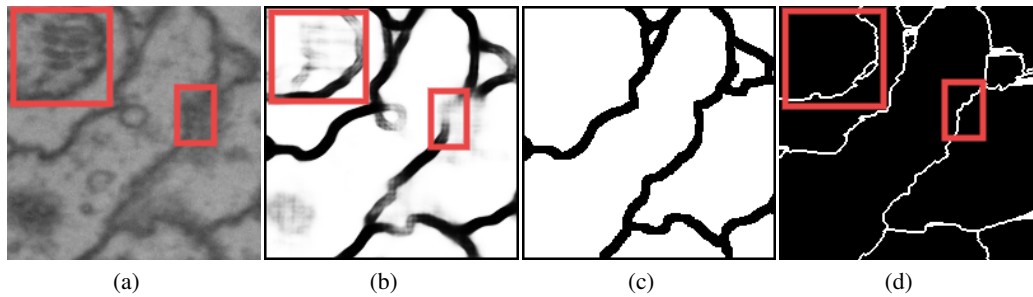

|      (a)      |      (b)      |      (c)      |      (d)      |

Figure 10: Illustration of two different types of topological errors captured by DMT-loss. **(a)** an input neuron image with challenging locations highlighted. **(b)** likelihood map of a baseline method without topological guarantee (Ronneberger et al., 2015). **(c)** ground truth. **(d)** The topologically critical structure from the likelihood, captured by the proposed discrete Morse algorithm. These structures will be used in the DMT-Loss.

DMT, analogous to classical Morse theory. We will however only briefly introduce some relevant concepts for the present paper, and we will describe it in the setting of cubical complexes (instead of simplicial complexes) as it is more suitable for images.

Let $K$ be a cubical complex. Given a $p$-cell $\tau$, we denote by $\sigma < \tau$ if $\sigma$ is a $(p-1)$-dimensional face for $\tau$. A *discrete gradient vector* (also called *a V-pair* for simplicity) is a pair $(\tau, \sigma)$ where $\sigma < \tau$. Now suppose we are given a collection of V-pairs $\mathsf{M}(K)$ over the cubical complex $K$. A sequence of cells $\pi : \tau_0^{p+1}, \sigma_1^p, \tau_1^{p+1}, \sigma_2^p, \cdots, \sigma_k^p, \tau_k^{p+1}, \sigma_{k+1}^p$, where the superscript $p$ in $\alpha^p$ stands for the dimension of this cell $\alpha$, form a *V-path* if $(\tau_i, \sigma_i) \in \mathsf{M}(K)$ for for any $i \in [1, k]$ and $\sigma_i < \tau_{i-1}$ for any $i \in [1, k+1]$. A V-path $\pi$ is *acyclic* if $(\tau_0, \sigma_{k+1}) \notin \mathsf{M}(K)$. This collection of V-pairs $\mathsf{M}(K)$ form a *discrete gradient vector field*[4] if (cond-i) each cell in $\mathsf{M}(K)$ can only appear in at most one pair in $\mathsf{M}(K)$; and (cond-ii) all V-paths in $\mathsf{M}(K)$ are acyclic. Given a discrete gradient vector field $\mathsf{M}(K)$, a simplex $\sigma \in K$ is *critical* if it is not in any V-pair in $\mathsf{M}(K)$.

Even though a discrete gradient vector (a V-pair), say $(\tau, \sigma)$ is a combinatorial pair instead of a real vector, it still indicates a "flow" from $\tau$ to its face $\sigma$. A V-path thus corresponds to a flow path (integral line) in the smooth setting. However, to make a collection of V-pairs a valid analog of gradient field, (cond-i) says that at each simplex there should only be one "flow" direction; while (cond-ii) is necessarily as flow lines traced by gradient can only go down in function values and thus never come back (thus acyclic).

---

[4]We will not introduce the concept of discrete Morse function, as the discrete gradient vector field is sufficient to define all relevation notations.

A critical simplex has "vanishing gradient" as it is not involved in any V-pair in $\mathsf{M}(K)$ (i.e, there is no flow at this simplex). Given a 2D cubical complex $K$ a a discrete gradient vector field $\mathsf{M}(K)$, we can view critical 0-, 1-, 2- and 3-cells as minima, saddle points, and maxima, respectively. If $K$ is 3D, then we can view critical 0-, 1-, 2- and 3-cells as minima, index-1 saddle, index-2 saddle and maxima, respectively.

Hence, a 1-stable manifold in 2D will correspond to a V-path connecting a critical square (a maximum) and a critical edge (a saddle), while in 3D, it will be a V-path connecting a critical cube and a critical square.

**Morse cancellation.** A given discrete gradient field $\mathsf{M}(K)$ could be noisy, e.g, there are shallow valleys where the mountain ridge around it should be ignored. Fortunately, the discrete Morse theory provides an elegant and purely combinatorial way to cancel pairs of critical simplices (and thus reduce their stable manifolds). In particular, given $\mathsf{M}(K)$, a pair of critical simplices $\langle \delta^{(p+1)}, \gamma^p \rangle$ is *cancellable* if there is a unique V-path $\pi = \delta = \delta_0, \gamma_1, \delta_1, \ldots, \delta_s, \gamma_{s+1} = \gamma$ from $\delta$ to $\gamma$. The *Morse cancellation operation* simple reverse all V-pairs along this path by removing all V-pairs along these path, and adding $(\delta_{i-1}, \gamma_i)$ to $\mathsf{M}(K)$ for any $i \in [1, s+1]$. It is easy to check that after the cancellation neither $\delta$ nor $\gamma$ is critical.

### A.2.2 PERSISTENCE PRUNING

We can extend this vertex-valued function $\rho$ to a function $\rho : K \to \mathbb{R}$, by setting $\rho(\sigma)$ for each cell to be the maximum $\rho$-value of each vertex in $\sigma$. How to obtain a discrete gradient vector field from such function $\rho : K \to \mathbb{R}$? Following the approach developed in (Wang et al., 2015; Dey et al., 2018), we initialize a trivial discrete gradient vector field where all cells are initially critical. Let $\epsilon > 0$ be a threshold for simplification. We then perform persistence algorithm (Edelsbrunner et al., 2000) induced by the super-level set filtration of $\rho$ and pair up all cells in $K$, denoted by $\mathcal{P}_\rho(K)$.

Persistent homology is one of the most important development in the field of topological data analysis in the past two decades (Edelsbrunner & Harer, 2010; Edelsbrunner et al., 2000; Zomorodian & Carlsson, 2005). We will not introduce it formally here. Imagine we grow the complex $K$ by starting from the empty set and gradually include more and more cells in decreasing $\rho$ values. (More formally, this is the so-called super-level set filtration of $K$ induced by $\rho$.) Through this course, new topological feature can be created upon adding a simplex $\sigma$, and sometiems a feature can be destroyed upon adding a simplex $\tau$. Persistence algorithm (Edelsbrunner et al., 2000) will pair up simplices; that is, its output is a set of pairs of simplices $\mathcal{P}_\rho(K) = \{(\sigma, \tau)\}$, where each pair captures the birth and death of topological features during this evolution. The persistence of a pair, say $\mathsf{p} = (\sigma, \tau)$, is defined as $\mathrm{pers}(\mathsf{p}) = \rho(\sigma) - \rho(\tau)$, measuring how long the topological feature captured by $\mathsf{p}$ lives in term of $\rho$. In this case, we also write $\mathrm{pers}(\sigma) = \mathrm{pers}(\tau) = \mathrm{pers}(\mathsf{p})$ – the persistence of a simplex (say $\sigma$ or $\tau$) can be viewed as the importance of this simplex.

With this intuition of the persistence pairings, we next perform Morse-cancellation operation to all pairs of these cells $(\sigma, \tau) \in \mathcal{P}_\rho(K)$ in increasing order their persistence if (i) its persistence $\mathrm{pers}(\delta, \gamma) < \epsilon$ (i.e, this pair has low persistence and thus not important); and (ii) this pair $(\delta, \gamma)$ is cancellable.

Let $\mathsf{M}_\epsilon(K)$ be the resulting discrete gradient field after simplifying all low-persistence critical simplices. We then construct the 1-stable and 2-stable manifolds for the remaining (high persistence, and thus important) saddles (critical 1-cell and 2-cells) from $\mathsf{M}_\epsilon(K)$. Let $\mathcal{S}_1(\epsilon)$ and $\mathcal{S}_2(\epsilon)$ be the resulting collection of 1- and 2-stable manifolds respectively. In particular, see an illustration of a V-path (highlighted in black) corresponding to a 1-stable manifold of the green saddle in Fig. 3(c).

### A.2.3 MORE DETAILS ON THE APPROXIMATION OF $\mathcal{S}_2$ VIA $\hat{\mathcal{S}}_2$.

We approximate $S_2$ by taking the boundary of the stable manifold of the minima (basins/valleys in the terrain). This is like a watershed algorithm: growing the basins from all minima until they meet. The stable manifolds of the minima is approximated using spanning trees. This algorithm is inspired by the continuous analog for Morse functions.

### A.3 EXPERIMENTS

### A.3.1 DATASETS

We conduct experiments on six different 2D datasets. The first three datasets are neuron images (Electron Microscopy images). The task is to segment membranes and eventually partition the image into neuron regions.

**CREMI** contains 125 images. The resolution for each image is 1250x1250.

**ISBI12** (Arganda-Carreras et al., 2015) contains 30 images. The resolution for each image is 512x512.

**ISBI13** (Arganda-Carreras et al., 2013) contains 100 images. The resolution for each image is 1024x1024.

The next three are natural image datasets, and their structures are vital for their functionality.

**CrackTree** (Zou et al., 2012) contains 206 images of cracks in road. The resolution for each image is 600x800.

**Road** (Mnih, 2013) has 1108 images from the Massachusetts Roads Dataset. The resolution for each image is 1500x1500.

**DRIVE** (Staal et al., 2004) is a retinal vessel segmentation dataset with 20 images. The resolution for each image is 584x565.

### A.3.2 EVALUATION METRICS

We use five different evaluation metrics to evaluate the proposed DMT-loss.

**Pixel-wise accuracy:** Pixel-wise accuracy is one of the most common metrics which measures the percentage of correctly classified pixels.

**DICE score:** DICE score (also known as DICE coefficient, DICE similarity index) is the same as the F1 score.

**Adapted Rand Index (ARI)**: ARI is the maximal F-score of the foreground-restricted Rand index, a measure of similarity between two clusters. On this version of the Rand index we exclude the zero component of the original labels (background pixels of the ground truth).

**Variation of Information (VOI)**: VOI is a measure of the distance between two clusterings. It is closely related to mutual information; indeed, it is a simple linear expression involving the mutual information.

**Betti number error**: which directly compares the topology (number of handles) between the segmentation and the ground truth. We randomly sample patches over the segmentation and report the average absolute difference between their Betti numbers and the corresponding ground truth patches.

### A.3.3 QUANTITATIVE RESULTS FOR MORE 2D DATASETS

Table. 3 shows quantitative results for ISBI12 and DRIVE.

### A.3.4 FAIRNESS COMPARISONS WITH SAME BACKBONE NETWORKS

We copy the numbers of TopoLoss from (Hu et al., 2019), which is TopoLoss+DIVE. And in this paper, we use U-Net as the backbone. Indeed, with U-Net, TopoLoss will be worse and the gap will be even bigger. The DIVE used in (Hu et al., 2019) is more expensive and better designed specifically for EM images. We choose U-Net in this manuscript as it is lightweight and easy to generalize to many datasets. We also apply our backbone-agnostic DMT-loss to the DIVE network (Fakhry et al., 2016). All the experiments are conducted on CREMI 2D dataset. The quantitative results (Betti Error) are shown in the Table 4.

### A.3.5 QUALITATIVE RESULTS FOR 3D DATASETS

Fig. 11 shows qualitative results for ISBI13 dataset.

Table 3: Quantitative results for different models on several 2D medical datasets

| Method | Accuracy | DICE | ARI | VOI | Betti Error |
|--------|----------|------|-----|-----|-------------|
| | | ISBI12 | | | |
| DIVE | $0.9640 \pm 0.0042$ | $0.9709 \pm 0.0029$ | $0.9434 \pm 0.0087$ | $1.235 \pm 0.025$ | $3.187 \pm 0.307$ |
| U-Net | $0.9678 \pm 0.0021$ | $0.9699 \pm 0.0048$ | $0.9338 \pm 0.0072$ | $1.367 \pm 0.031$ | $2.785 \pm 0.269$ |
| Mosin. | $0.9532 \pm 0.0063$ | $0.9716 \pm 0.0022$ | $0.9312 \pm 0.0052$ | $0.983 \pm 0.035$ | $1.238 \pm 0.251$ |
| TopoLoss | $0.9626 \pm 0.0038$ | $0.9755 \pm 0.0041$ | $0.9444 \pm 0.0076$ | $0.782 \pm 0.019$ | $\mathbf{0.429 \pm 0.104}$ |
| DMT | $0.9593 \pm 0.0035$ | $\mathbf{0.9796 \pm 0.0033}$ | $\mathbf{0.9527 \pm 0.0052}$ | $\mathbf{0.671 \pm 0.027}$ | $\mathbf{0.391 \pm 0.114}$ |
| | | DRIVE | | | |
| DIVE | $0.9549 \pm 0.0023$ | $0.7543 \pm 0.0008$ | $0.8407 \pm 0.0257$ | $1.936 \pm 0.127$ | $3.276 \pm 0.642$ |
| U-Net | $0.9452 \pm 0.0058$ | $0.7491 \pm 0.0027$ | $0.8343 \pm 0.0413$ | $1.975 \pm 0.046$ | $3.643 \pm 0.536$ |
| Mosin. | $0.9543 \pm 0.0047$ | $0.7218 \pm 0.0013$ | $0.8870 \pm 0.0386$ | $1.167 \pm 0.026$ | $2.784 \pm 0.293$ |
| TopoLoss | $0.9521 \pm 0.0042$ | $0.7621 \pm 0.0036$ | $0.9024 \pm 0.0113$ | $1.083 \pm 0.006$ | $\mathbf{1.076 \pm 0.265}$ |
| DMT | $0.9495 \pm 0.0036$ | $\mathbf{0.7733 \pm 0.0039}$ | $\mathbf{0.9077 \pm 0.0021}$ | $\mathbf{0.876 \pm 0.038}$ | $\mathbf{0.873 \pm 0.402}$ |

Table 4: Comparison with same backbones

| | U-Net | DIVE |
|--------|-------|------|
| TopoLoss | $1.451 \pm 0.216$ | $1.113 \pm 0.224$ |
| DMT-Loss | $\mathbf{0.982 \pm 0.179}$ | $\mathbf{0.956 \pm 0.142}$ |

### A.3.6 THE ABLATION STUDY FOR BALANCED TERM $\beta$.

We conduct another ablation study for the balanced weight of parameter $\beta$. Note that, the parameter $\beta$ is dataset dependent. We conduct the ablation experiment on CREMI 2D dataset. When $\beta = 3$, the proposed DMT-loss achieves best performance 0.982 (Betti Error). When $\beta = 2$ and $\beta = 4$, the performance drops to 1.074 and 1.181 (both in Betti Error), respectively. The parameter $\beta$ is also chosen via cross-validation.

### A.3.7 COMPARISON WITH OTHER SIMPLER CHOICES

Table 5 shows the results comparing with Canny edge detection and ridge detection

Table 5: Comparison with other simpler choices

| Method | Accuracy | Betti Error |
|--------|----------|-------------|
| DMT | 0.9475 | 0.982 |
| Canny edge detection | 0.9386 | 2.971 |
| Ridge detection | 0.9443 | 2.507 |

### A.3.8 RESULTS WITH DIFFERENT NOISE LEVELS

In Table 6, 10% means the percentage of corrupted pixels, and $\delta/2\delta$ means the sdv of the added Gaussian noise. For reference, we note that the performance of the standard U-Net is 3.016 (Betti Error).

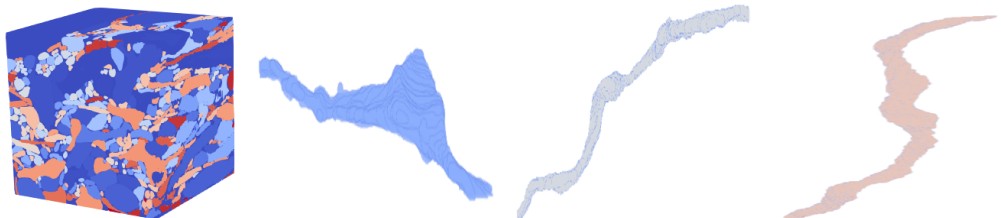

Figure 11: Segmentation results for ISBI13 dataset and 3 randomly selected neurons.

Table 6: Results with different noise levels

| Method | Accuracy | Betti Error |
|---|---|---|
| DMT | 0.9475 | 0.982 |
| Gaussian $(10\%, \delta)$ | 0.9393 | 1.086 |
| Gaussian $(20\%, \delta)$ | 0.9272 | 1.391 |

| Method | Accuracy | Betti Error |
|---|---|---|
| DMT | 0.9475 | 0.982 |
| Reweighted CE | 0.9481 | 2.753 |

