# OpenReview forum: "Topology-Aware Segmentation Using Discrete Morse Theory"
_ICLR.cc/2021/Conference — ICLR 2021 Spotlight_

### Official Review · AnonReviewer3 · 2020-10-26
**Improvement of Topoloss**

**Rating:** 6
**Confidence:** 4

**Review:**

The paper introduces an elegant, yet not fully novel, idea: use a discrete morse complex to define a new loss for image and volume segmentation. The idea of using topology in the loss is similar to TopoLoss, but the use of larger critical structures makes this approach slightly outperform topoloss.

The paper is in a difficult spot: most of the paper (pages 2-6) provides an overview of Morse theory, the discrete version, and an algorithm to compute it on images. I don’t see any novelty there, as it seems to be all coming from the cited papers, except for the loss in section 2.3. Could you clarify if this is the case or if there is anything else that is novel? If indeed that is the only novelty the paper could be shortened by citing the corresponding papers instead of repeating well-known material  (or maybe moving it to an appendix to make it self-contained).

The comparison is extensive and while the method outperforms most of the baselines, the results are quite similar (2D) to TopoLoss (which uses a loss based on persistent homology, closely related to the one proposed here), with minor differences (the more noticeable in the betti error, which is ~10-15% better). In 3D the difference is even smaller and there is no clear winner between the two.

I do not understand how the Morse structure used in the loss can be differentiable. The authors state explicitly that both the structure and the mask change not continuously. I don’t understand the logical step that leads to the conclusion that the loss is differentiable, could you clarify?

Overall, the paper is a variant of Topoloss which has slightly better results. I am neutral on this paper as I am not familiar with the expectations for an ICLR paper. I don’t think the paper novelty is sufficient for acceptance, but there are measurable numerical improvements in the 2D results that might be useful in applications (no applications of the resulting segmentations are shown in the paper, unfortunately).

---

> ### Author Response · Authors · 2020-11-18
> **Novellty, improvement over TopoLoss and differentiability**
>
> We thank the reviewer for the thorough review and comments. We will improve the presentation according to the suggestions. Below we address specific concerns one-by-one.
>
> **Q1**: Using existing discrete Morse theory is not novel.
> **A**: We do not think this paper is incremental just because we are using an existing mathematical theory. Our major novelty is the application of this principled mathematical tool to the topology-aware segmentation problem in an effective manner, which requires deep insights into both the method and the problem to devise an effective loss as ours.
>
> Technically, our algorithm is nontrivial. We proposed to (1) prune false-positive Morse structures using persistent homology and Morse cancellation; (2) linear-time approximation of 2D Morse structures based on duality. Both contributions came from numerous trials-and-errors and are well justified in practice. We consider our contributions valuable to not only practitioners, but also theoreticians. These contributions were listed in the introduction, and further expanded in Section 2.2.
>
> Conceptually, it is not incremental to leap from topoloss to DMT-loss, from critical pixels to critical structures. Both topological loss and topology-aware segmentation are fairly new and not perfect. Improving the methodology requires  customizing  theoretical tools to better solve the practical problem. These kinds of contributions, although not fitting for a theoretical venue, are  well-justified for conferences that are more problem-oriented, such as ICLR.
>
> **Q2**: Most of the paper (pages 2-6) provides an overview of Morse theory, the discrete version, and an algorithm to compute it on images. The paper could be shortened by citing the corresponding papers instead of repeating well-known material (or maybe moving it to an appendix to make it self-contained).
> **A**: Thanks for the suggestion. We note that Morse theory is not well-known in the machine learning and computer vision community, besides people who work on computational topology. Hence, we believe an introduction is necessary. We have tried hard to condense the background to about 1.5 pages (page 3 middle to page 4). Pages 5-6 explain nontrivial technical contributions (pruning and approximating) and the loss function.  We believe this provides important intuition to practitioners (most of whom are not familiar with this subject), so that they will not treat the method as a blackbox. The transparency will help the method, and in a broader sense topological approaches, to be better accepted and adapted in practice.
>
> **Q3**: The empirical improvement over Topoloss is minor.
> **A**: The major benefits of DMT-loss over TopoLoss are twofold: (1) improved topological accuracy; and (2) better training efficiency. The first benefit is clearly demonstrated on 3D images (Table 2). The Betti error performance is improved by up to 37%. On 2D images, DMT-loss is better than TopoLoss, but not as significantly as in 3D images. In terms of the training efficiency, the speedup is significant. For 2D CREMI dataset, DMT-loss is much faster to train than Topoloss (1.2h vs 3h, reported in Page 8). For 3D images, TopoLoss can take half a day to train, whereas DMT-loss only needs 3 to 4 hours.
>
> **Q4**: I do not understand how the Morse structure used in the loss can be differentiable.
> **A**: Thanks for pointing out the confusion. The computed discrete Morse structures are uniquely determined by the ordering of pixels with regard to the likelihood function. We assume a general setting in which all pixels have distinct likelihood function values. Within a sufficiently small neighborhood of the likelihood function, the ordering of all pixels remains unchanged. Thus the discrete Morse structures $M_f$ remain constant, and the loss $L_{dmt}$ is differentiable. This argument is similar to the differentiability argument of TopoLoss. We will clarify in the final version.

---

### Official Review · AnonReviewer1 · 2020-10-28
**Interesting method, but missing details and comparison to state of the art segmentation methods**

**Rating:** 5
**Confidence:** 3

**Review:**

## Summary and contributions
This paper tackles the problem of topological correctness of segmentation of fine structures.
Especially for the study of biological or medical processes this is often essential.
The authors propose to use discrete Morse theory (DMT) to help a deep neural network focus on areas that are currently topologically incorrectly segmented.
The result of DMT is similar to the classical watershed transform, yet provides a sound mathematical basis.
Moreover the paper claims that commonly used metrics often do not value topological correctness enough and proposes to use metrics that are more in line with this goal.

## Strengths
- The paper combines theoretical advances in topology and Morse theory with current practical methods
 in deep learning. Thus it tries to motivate its approach with a sound mathematical basis.
- The imbalances often present in datasets with fine structures, resulting in a sampling bias, are are alleviated using global features.
- It is written in a clear and comprehensible way
- The method is evaluated on multiple datasets and compared to a range of related work

## Weaknesses:
- Many details are missing to make this work reproducible
- The method uses topology to create a mask for the loss, however the topology is not directly enforced in the loss
- Even though most of the datasets are from challenges, the results are not compared to the current leader board


## Clarity:
For the most parts the paper is written very clearly. The introductions to Morse theory are well understandable.
The related work part is a little dense, making it hard to extract really relevant work.


## Reproducibility:
In the current form it is not possible to reproduce the results without considerable effort.
There are many parameters missing.
- It is mentioned that epsilon is chosen using cross-validation, but not in which value this resulted, the same holds for beta.
- The authors mention that a neural network architecture inspired by the U-Net is used, however all details about the specifics of the used networks are missing (dimensions, layers, feature maps, learning rate, used optimizer etc)
- The authors mention required training time, however without specifying the used hardware or number of steps.
- Will the code be published? Is existing code used to compute the Morse structures or did the authors implement it?


## Additional feedback, comments, suggestions for improvement
General comments:
- It is a nice aspect for the reader that the authors tried to distinguish significant from insignificant improvements over previous methods.

Requests for clarification:
- Could the authors please clarify how they use the word epoch? Usually it is used to describe one pass through the whole dataset, but it sounds like
it is used here to refer to a single step.
- Page 5, computation: Could the authors please clarify how the (costly) persistence computation can be avoided if it is needed, according to the next
sentence, to distinguish between below and above the persistence threshold epsilon.
- The result seems to be similar to an increased weight for FP,FN and TP pixels or a decreased weight for TN pixels.
Could the authors please elaborate on how the effect of their method differs from this.
Have the authors performed experiments with this as a baseline/part of an ablation study (with cross-validated weights)?
This would be very interesting.

Claims that might be formulated a bit to strongly:
- The authors correctly claim that unbalanced data is an issue and that just using more annotated data (with the same unbalance) is often not a simple
remedy. This is often true, however, a number of approaches have been developed to alleviate this problem and more annotations with more hard cases
can be helpful for it.
- The authors claim that "by focusing on more critical locations early, our method is more likely to escape poor local minima of the loss landscape".
Can the authors please elaborate on the experimental basis for this claim?
- The authors claim, citing related work (Hardt et al), that shorter training time might improve performance. However, just the same, there is related
work claiming the opposite ([1]). This seems to be an open problem and thus speculating about it should be founded in experimental data.
- Fig 1, caption ("baseline ...without topological guarantee"): This gives the impression that the presented method provides topological guarantees.
In the introduction the authors mention that the method tackles the sampling bias, which sounds reasonable, but it does not provide guarantees. If it
does provide guarantees, could the authors please elaborate on how these are strongly enforced?

Suggestions:
- The authors nicely cite a lot of related work. However, it would be interesting how they are different and how the work at hand tackles specific
problems that are left open by them. Or if they solve the same problem with a very different approach, which is also perfectly fine and still valuable
for the field.
- The complexity given for 3d images for S_1 gives, to a cursory look, the slightly misleading impression that it is still n log n whereas it seems to
be more than quadratic due to the n^w in T.
- The conclusion might benefit from a few more details regarding for example remaining issues.
- I understand that there is a page limit, however the discussion of the results comes up a bit short.

Typos:
- Page 4, Sec 2.2: on the the original definition

## note about review:
- I cannot fully judge the formal correctness of the authors' description of DMT.

[1] Hoffer et. al.: Train longer, generalize better: closing thegeneralization gap in large batch training of neural networks

---

> ### Author Response · Authors · 2020-11-18
> **Response to the comments of Reviewer #1 (1/2)**
>
> We thank the reviewer for the thorough review and comments. We will improve the presentation according to the suggestions. Below we address specific concerns one-by-one.
>
> **Q1**: The result of DMT is similar to the classical watershed transform, yet provides a sound mathematical basis.
> **A**: The usage of DMT is essential. DMT covers a much more general setting than watershed. For example, DMT can recover 1D skeletons in 3D images (vessels), whereas watershed cannot. Furthermore, pruning of spurious structures is essential to success. DMT structures can be conveniently pruned by persistent homology. Watershed is usually pruned by thresholds (saddle point values), which are not ideal for identifying topology-critical structures.
>
> *(Weakness)*
> **Q2**: The method uses topology to create a mask for the loss, however the topology is not directly enforced in the loss.
> **A**: The Morse structures capture the topology of the predicted likelihood map. By comparing these structures with ground truth masks, we identify and penalize the topology-critical errors of the prediction.
>
> **Q3**: Even though most of the datasets are from challenges, the results are not compared to the current leader board.
> **A**: In order to validate the efficacy of the proposed loss, we applied it to a broad spectrum of datasets using a popular light-weight backbone model (U-Net). In principle, our loss can be combined with any neural network, including those heavy networks that are optimized for specific data/competition. We expect our loss will improve the topological accuracy of these models as well. But these data-specific methods are out of the scope of this paper.
>
> **Q4**: For the most part the paper is written very clearly. The introductions to Morse theory are well understandable. The related work part is a little dense, making it hard to extract really relevant work.
> **A**: Thanks for your suggestion. We will improve the manuscript by adding more details into the related work section.
>
> *(Reproducibility)*
> **Q5**: Actual values of the hyperparameters epsilon and beta.
> **A**: $\epsilon$ and $\beta$ are data dependent. Our ablation study on the CREMI 2D dataset (page 8 and page 17) showed the sensitivity of the parameters $\epsilon$ and $\beta$. We also mentioned that for this dataset, we select $\epsilon$ = 0.2 and $\beta$ = 3.
>
> **Q6**: Specifics of the U-Net (dimensions, layers, feature maps, learning rate, used optimizer etc).
> **A**: We would like to emphasize that our DMT-loss is model agnostic, and it could be incorporated into any backbones/segmenter to improve its topological accuracy. The 2D U-Net in this paper consists of two 3x3 convolutions, each followed by a ReLU and a 2x2 max pooling operation. At each downsampling step the number of feature channels is doubled. In the expansive path, each step consists of an upsampling of the feature map followed by a 2x2 convolution that halves the number of feature channels, a concatenation with the correspondingly cropped feature map from the contracting path, and two 3x3 convolutions, each followed by a ReLU. We implement our model with PyTorch and apply RMSprop optimizer with a learning rate of $10^{-4}$.
>
> **Q7**: The used hardware or number of steps.
> **A**: All the methods are tested on a PC with two Intel E5-2650 2.20GHz CPU and one GeForce GTX 1080Ti.
>
> **Q8**: Will the code be published? Is existing code used to compute the Morse structures or did the authors implement it?
> **A**: We implemented the code for approximating Morse structures. The codes will be released.
>
> *(Additional feedback -- clarification)*
> **Q9**: Could the authors please clarify how they use the word epoch? Usually it is used to describe one pass through the whole dataset, but it sounds like it is used here to refer to a single step.
> **A**: In each epoch, we train the model through all training images once. For each training image, aside from cross-entropy loss, we also evaluate the DMT-loss and its gradient for backpropagation. We’ll clarify this in the final version.
>
> **Q10**: Could the authors please clarify how the (costly) persistence computation can be avoided.
> **A**: Persistence computation cannot be avoided. Indeed, the computation time of DMT loss/gradient for each iteration is similar to TopoLoss. The training efficiency of DMT-loss comes from faster convergence rate compared with TopoLoss. The number of epochs for DMT-loss to converge is only about half of that of TopoLoss. More details can be found in Page 8, the paragraph titled “The benefit of the proposed DMT-loss”.

---

> > ### Author Response · Authors · 2020-11-18
> > **Response to the comments of Reviewer #1 (2/2)**
> >
> > **Q11**: The result seems to be similar to an increased weight for FP,FN and TP pixels or a decreased weight for TN pixels. Could the authors please elaborate on how the effect of their method differs from this. Have the authors performed experiments with this as a baseline/part of an ablation study (with cross-validated weights)? This would be very interesting.
> > **A**: Good suggestion! We run an additional ablation study to compare with a baseline of reweighting the FP and FN pixels in cross-entropy loss (Reweighting CE). The weights of the FP/FN pixels are hyperparameters tuned via cross-validation. As shown in the table below (on CREMI 2D data), DMT-loss is better than this baseline. The reason is that the DMT-loss specifically penalizes FP and FN pixels which are topology-critical. Meanwhile, reweighting CE adds weights to FP/FN pixels without discrimination. A majority of these misclassified pixels are not topology-critical. They are near the boundary of the foreground. We added a figure to illustrate this (see Page 18, Figure 11 of the updated paper).
> >
> > |Method|Accuracy|Betti Error|
> > |----|----|----|
> > |DMT|0.9475|0.982|
> > |Reweighing CE |0.9481|2.753 |
> >
> > **Q12**: A number of approaches have been developed to alleviate unbalanced data and more annotations with more hard cases can be helpful for it.
> > **A**: Existing hard sample mining methods are not explicitly targeting topology-critical pixels. Thus they cannot be very effective in improving the topological accuracy of the model. As illustrated in the previous question, many misclassified pixels are not topology-critical. Even if we carry out an interactive algorithm to annotate hard cases, a topological algorithm (persistence or DMT) is necessary to discover the topology-critical ones for annotation. This will be left as a future work.
> >
> > **Q13**: The authors claim that "by focusing on more critical locations early, our method is more likely to escape poor local minima of the loss landscape". Can the authors please elaborate on the experimental basis for this claim?
> > **A**: To show that DMT-loss leads to a better local optimum, we compare the quality of the model trained using topoloss (Topo-model) and the model trained using DMT-loss (DMT-model). We evaluate both models on the topoloss (cross-entropy + persistent-homology-based loss). On the CREMI 2D dataset, the topo-model achieves 0.107 loss whereas the DMT-model achieves 0.083 loss. In other words, training with DMT-loss results in a better local optimum on the loss landscape of topoloss. This is also reflected on the generalization performance. We have shown that models trained with DMT-loss achieve better validation performance.
> >
> > **Q14**: The claim “The shorter training time may also contribute to better stability of the SGD algorithm, and thus better test accuracy (Hardt et al., 2016).” is debatable.
> > **A**: Thanks. We will tune down the hypothesis in the final version.
> >
> > **Q15**: The complexity given for 3d images for $S_1$ gives, to a cursory look, the slightly misleading impression that it is still $nlogn$ whereas it seems to be more than quadratic due to the $n^w$ in $T$.
> > **A**: We gave the formula of $T$ ($T=n^\omega$) right after the complexity formula. Since they are very closeby, we think readers will not misunderstand.

---

> > ### Comment · ~Qin_Liu3 · 2021-05-24
> > **Release of the code**
> >
> > The authors replied the reviewer that they would release the code, but six months later we still cannot find the release.

---

> > > ### Comment · ~Xiaoling_Hu1 · 2021-05-24
> > > **Github link**
> > >
> > > Here is the github link: https://github.com/HuXiaoling/DMT_loss.

---

### Official Review · AnonReviewer4 · 2020-10-28
**An interesting piece of theory**

**Rating:** 8
**Confidence:** 3

**Review:**

This manuscript introduces a segmentation framework that combines discrete morse theory to encourage topological correctness of the segmentation. Based on the evaluation on both 2D and 3D datasets, the proposed method has the potential to improve segmentation in selective applications where the topology is of great interest.

Some constructive feedbacks:
While the Morse theory itself seems to be interesting, the way it is incorporated with deep learning can be better justified. The MST algorithm is not a functional, is non-deterministic, and changes with iterations, so the converging property of the objective function is not clearly studied.

Both hyperparameters are not chosen by inner cross-validation. Instead, the optimal results reported for the proposed method are based on a search via cross-validation, which is not the case for other baselines according to my reading.

The proposed method essentially highlights segmentation accuracy in geometrically rich regions. I wonder how essential it is to have the Morse theory compared to other simpler choices, e.g. edge detector, ridges/valleys estimator, or even just a gradient field.

Often times the biggest drawbacks for conventional geometric methods are their inferior generalizability and vulnerability to noise. An experiment with respect to noise level can be very helpful

---

> ### Author Response · Authors · 2020-11-18
> **New ablation studies w.r.t. baselines and noise**
>
> We thank the reviewer for the thorough review and comments. We will improve the presentation according to the suggestions. Below we address specific concerns one-by-one.
>
> **Q1**: DMT algorithm is not a functional, is non-deterministic, and changes with iterations, so the converging property of the objective function is not clearly studied.
> **A**: Good question! Morse structures do change as the likelihood function changes. Empirically, we have observed that the loss behaves well. The likelihood function tends to converge, and the Morse structures also converge. Since DMT-loss only affects a small set of pixels, it might be the case that the converging behavior of the cross-entropy loss dominates the training. A theoretical analysis of the behavior of the loss will be left as future work.
>
> **Q2**: Both hyperparameters are not chosen by inner cross-validation. Instead, the optimal results reported for the proposed method are based on a search via cross-validation, which is not the case for other baselines according to my reading.
> **A**: As mentioned in Sec 3 Paragraph 1 (Page 6), we use a 3-fold cross-validation to tune hyperparameters for both the proposed method and other baselines. This is a fair comparison.
>
> **Q3**: I wonder how essential it is to have the Morse theory compared to other simpler choices, e.g. edge detector, ridges/valleys estimator, or even just a gradient field.
> **A**: We compare with two baselines: canny edge detection, and ridge detection. The results on the CREMI 2D dataset are reported below. Although our focus is Betti error, we also report per-pixel accuracy for reference. From the results we observe that the baseline models couldn't solve topological errors, even though they achieve high per-pixel accuracy. Without a persistent-homology based pruning, these baselines generate too many noisy structures, and thus are not as effective as DMT-loss.
>
> |Method|Accuracy|Betti Error|
> |----|----|----|
> |DMT|0.9475|0.982|
> |Canny edge detection|0.9386|2.971|
> |Ridge detection|0.9443|2.507|
>
> **Q4**: An experiment with respect to noise level can be very helpful.
> **A**: This is a good suggestion. We run an ablation study on images corrupted with Gaussian noise. The experiment is also conducted on the CREMI 2D dataset. In the table, 10% means the percentage of corrupted pixels, and $\delta/2\delta$ means the sdv of the added Gaussian noise. DMT-loss is fairly robust and remains of good performance even with high noise levels. The reason is that the Morse structures are computed on the likelihood map, which is already robust to noise. For reference, we note that the performance of the standard U-Net is 3.016 (Betti Error).
>
> |Noise level|Accuracy|Betti Error|
> |----|----|----|
> |DMT (w/o noise)|0.9475|0.982|
> |Gaussian (10%, $\delta$) |0.9393|1.086|
> |Gaussian (10%,  $2\delta$)|0.9272|1.391|

---

> > ### Comment · AnonReviewer4 · 2020-11-18
> > **New results are promising**
> >
> > Given the new results of Q3 & Q4, the manuscript is stronger so I raise my evaluation score. New results should be included in the supplement.
> >
> > Q2: Sec 3 Paragraph 1 (Page 6) does not say anything about hyperparameter tuning. The authors mentioned "$\epsilon$ and $\beta$are chosen via cross-validation", but did not explicitly say how they tune the parameters of the baselines. E.g., TopoLoss (Hu 2019) also has a balancing parameter. Please clearly indicate how the parameters of the baselines were chosen in the revised manuscript, or acknowledge this as a limitation.

---

> > > ### Author Response · Authors · 2020-11-19
> > > **Clarify the hyperparameter tuning**
> > >
> > > Thanks for your positive feedback and suggestions!
> > > In Sec Paragraph 1 (Page 6) of the updated version paper, we add a new sentence 'For all the experiments, we use a 3-fold cross-validation to tune hyperparameters for both the proposed method and other baselines, and report the mean performance over the validation set.' to clarify your concern.
> > > If you still have any concerns, please let us know. We're always willing to clarify any concern and improve the paper quality based on your valuable comments.

---

### Official Review · AnonReviewer2 · 2020-10-28
**Interesting and well-grounded method for end-to-end learning of topologically correct segmentations; evaluated on a large number of benchmark datasets**

**Rating:** 7
**Confidence:** 2

**Review:**

The paper focus on the segmentation of images in which the correct topology of the segmentation an important role plays. The key idea of the paper is the use of Discrete Morse Theory to identify those areas of the segmentation/likelihood map that are important for ensuring the correct topology of the segmentation. This is achieved by modulating the normal binary cross entropy loss function so that areas of Morse structures are focused on during the learning.

While DMT  is not my area of core expertise, I believe that the proposed use Morse structures for the end-to-end training of a network is novel and a very interesting idea. While the basic idea sounds simple, the implementation is not that simple. In particular, the computation of Morse structures is slow and also sometimes unstable. The paper addresses all of these issues nicely (adopting a fast approximation and pruning step).

The paper is well written but still difficult to read for somebody who is not very familiar with DMT. The proposed approach has been well evaluated on a number of biomedical and non-biomedical datasets. This evaluation includes the comparison to other approaches with and without topology-aware segmentation.

---

> ### Author Response · Authors · 2020-11-18
> **Response to the comments of Reviewer #2**
>
> **Q**: The paper is well written but still difficult to read for somebody who is not very familiar with DMT.
> **A**: Thanks for your positive feedback and valuable suggestions! We will improve our paper for better understanding, especially for non-expert readers.

---

### Decision · Program_Chairs · 2021-01-07
**Final Decision**

**Decision:**

Accept (Spotlight)

**Comment:**

The paper proposes a novel loss for segmentation tasks, which incorporates reasoning about topological accuracy of predicted segmentations vs. ground truth. All reviewers, after the rebuttal period, recommend acceptance, and I agree -- it's an interesting paper, offering a potentially useful and clearly novel building block for training segmentation models.